# Scalable Influence Estimation in Continuous-Time Diffusion Networks

**Nan Du**\*     **Le Song**\*     **Manuel Gomez-Rodriguez**[†]     **Hongyuan Zha**\*

Georgia Institute of Technology\*     MPI for Intelligent Systems[†]

dunan@gatech.edu     lsong@cc.gatech.edu
manuelgr@tue.mpg.de     zha@cc.gatech.edu

## Abstract

If a piece of information is released from a media site, can we predict whether it may spread to one million web pages, in a month ? This influence estimation problem is very challenging since both the time-sensitive nature of the task and the requirement of scalability need to be addressed simultaneously. In this paper, we propose a randomized algorithm for influence estimation in continuous-time diffusion networks. Our algorithm can estimate the influence of every node in a network with $|\mathcal{V}|$ nodes and $|\mathcal{E}|$ edges to an accuracy of $\epsilon$ using $n = O(1/\epsilon^2)$ randomizations and up to logarithmic factors $O(n|\mathcal{E}|+n|\mathcal{V}|)$ computations. When used as a subroutine in a greedy influence maximization approach, our proposed algorithm is guaranteed to find a set of $C$ nodes with the influence of at least $(1 - 1/e)\,\mathrm{OPT} -2C\epsilon$, where OPT is the optimal value. Experiments on both synthetic and real-world data show that the proposed algorithm can easily scale up to networks of millions of nodes while significantly improves over previous state-of-the-arts in terms of the accuracy of the estimated influence and the quality of the selected nodes in maximizing the influence.

## 1 Introduction

Motivated by applications in viral marketing [1], researchers have been studying the influence maximization problem: find a set of nodes whose initial adoptions of certain idea or product can trigger, *in a time window*, the largest expected number of follow-ups. For this purpose, it is essential to accurately and efficiently estimate the number of follow-ups of an arbitrary set of source nodes within the given time window. This is a challenging problem for that we need first accurately model the timing information in cascade data and then design a scalable algorithm to deal with large real-world networks. Most previous work in the literature tackled the influence estimation and maximization problems for infinite time window [1, 2, 3, 4, 5, 6]. However, in most cases, influence must be estimated or maximized up to a given time, *i.e.*, a finite time window must be considered [7]. For example, a marketer would like to have her advertisement viewed by a million people in one month, rather than in one hundred years. Such time-sensitive requirement renders those algorithms which only consider static information, such as network topologies, inappropriate in this context.

A sequence of recent work has argued that modeling cascade data and information diffusion using *continuous-time* diffusion networks can provide significantly more accurate models than discrete-time models [8, 9, 10, 11, 12, 13, 14, 15]. There is a twofold rationale behind this modeling choice. First, since follow-ups occur asynchronously, continuous variables seem more appropriate to represent them. Artificially discretizing the time axis into bins introduces additional tuning parameters, like the bin size, which are not easy to choose optimally. Second, discrete time models can only describe transmission times which obey an exponential density, and hence can be too restricted to capture the rich temporal dynamics in the data. Extensive experimental comparisons on both synthetic and real world data showed that continuous-time models yield significant improvement in settings such as recovering hidden diffusion network structures from cascade data [8, 10] and predicting the timings of future events [9, 14].

However, estimating and maximizing influence based on continuous-time diffusion models also entail many challenges. First, the influence estimation problem in this setting is a difficult graphical model inference problem, *i.e.*, computing the marginal density of continuous variables in loopy graphical models. The exact answer can be computed only for very special cases. For example, Gomez-Rodriguez et al. [7] have shown that the problem can be solved exactly when the transmission functions are exponential densities, by using continuous time Markov processes theory. However, the computational complexity of such approach, in general, scales exponentially with the size and density of the network. Moreover, extending the approach to deal with arbitrary transmission functions would require additional nontrivial approximations which would increase even more the computational complexity. Second, it is unclear how to scale up influence estimation and maximization algorithms based on continuous-time diffusion models to millions of nodes. Especially in the maximization case, even a naive sampling algorithm for approximate inference is not scalable: $n$ sampling rounds need to be carried out for each node to estimate the influence, which results in an overall $O(n|\mathcal{V}||\mathcal{E}|)$ algorithm. Thus, our goal is to design a scalable algorithm which can perform influence estimation and maximization in this regime of networks with millions of nodes.

In particular, we propose CONTINEST (**Con**tinous-**T**ime **In**fluence **Est**imation), a scalable randomized algorithm for influence estimation in a continuous-time diffusion network with heterogeneous edge transmission functions. The key idea of the algorithm is to view the problem from the perspective of graphical model inference, and reduces the problem to a neighborhood estimation problem in graphs. Our algorithm can estimate the influence of every node in a network with $|\mathcal{V}|$ nodes and $|\mathcal{E}|$ edges to an accuracy of $\epsilon$ using $n = O(1/\epsilon^2)$ randomizations and up to logarithmic factors $O(n|\mathcal{E}| + n|\mathcal{V}|)$ computations. When used as a subroutine in a greedy influence maximization algorithm, our proposed algorithm is guaranteed to find a set of nodes with an influence of at least $(1 - 1/e)$ OPT $-2C\epsilon$, where OPT is the optimal value. Finally, we validate CONTINEST on both influence estimation and maximization problems over large synthetic and real world datasets. In terms of influence estimation, CONTINEST is much closer to the true influence and much faster than other state-of-the-art methods. With respect to the influence maximization, CONTINEST allows us to find a set of sources with greater influence than other state-of-the-art methods.

## 2 Continuous-Time Diffusion Networks

First, we revisit the continuous-time generative model for cascade data in social networks introduced in [10]. The model associates each edge $j \rightarrow i$ with a transmission function, $f_{ji}(\tau_{ji})$, a density over time, in contrast to previous discrete-time models which associate each edge with a fixed infection probability [1]. Moreover, it also differs from discrete-time models in the sense that events in a cascade are not generated iteratively in rounds, but event timings are sampled directly from the transmission function in the continuous-time model.

**Continuous-Time Independent Cascade Model.** Given a *directed* contact network, $\mathcal{G} = (\mathcal{V}, \mathcal{E})$, we use a continuous-time independent cascade model for modeling a diffusion process [10]. The process begins with a set of infected source nodes, $\mathcal{A}$, initially adopting certain *contagion* (idea, meme or product) at time zero. The contagion is transmitted from the sources along their out-going edges to their direct neighbors. Each transmission through an edge entails a *random* spreading time, $\tau$, drawn from a density over time, $f_{ji}(\tau)$. We assume transmission times are independent and possibly distributed differently across edges. Then, the infected neighbors transmit the contagion to their respective neighbors, and the process continues. We assume that an infected node remains infected for the entire diffusion process. Thus, if a node $i$ is infected by multiple neighbors, only the neighbor that first infects node $i$ will be the *true parent*. As a result, although the contact network can be an arbitrary directed network, each cascade (a vector of event timing information from the spread of a contagion) induces a Directed Acyclic Graph (DAG).

**Heterogeneous Transmission Functions.** Formally, the transmission function $f_{ji}(t_i|t_j)$ for directed edge $j \rightarrow i$ is the conditional density of node $i$ getting infected at time $t_i$ given that node $j$ was infected at time $t_j$. We assume it is shift invariant: $f_{ji}(t_i|t_j) = f_{ji}(\tau_{ji})$, where $\tau_{ji} := t_i - t_j$, and nonnegative: $f_{ji}(\tau_{ji}) = 0$ if $\tau_{ji} < 0$. Both parametric transmission functions, such as the exponential and Rayleigh function [10, 16], and nonparametric function [8] can be used and estimated from cascade data (see Appendix A for more details).

**Shortest-Path property.** The independent cascade model has a useful property we will use later: given a sample of transmission times of all edges, the time $t_i$ taken to infect a node $i$ is the length

of the shortest path in $\mathcal{G}$ from the sources to node $i$, where the edge weights correspond to the associated transmission times.

## 3   Graphical Model Perspectives for Continuous-Time Diffusion Networks

The continuous-time independent cascade model is essentially a directed graphical model for a set of *dependent* random variables, the infection times $t_i$ of the nodes, where the conditional independence structure is supported on the contact network $\mathcal{G}$ (see Appendix B for more details). More formally, the joint density of $\{t_i\}_{i \in \mathcal{V}}$ can be expressed as

$$p\left(\{t_i\}_{i \in \mathcal{V}}\right) = \prod_{i \in \mathcal{V}} p\left(t_i | \{t_j\}_{j \in \pi_i}\right), \tag{1}$$

where $\pi_i$ denotes the set of parents of node $i$ in a cascade-induced DAG, and $p(t_i | \{t_j\}_{j \in \pi_i})$ is the conditional density of infection $t_i$ at node $i$ given the infection times of its parents.

Instead of directly modeling the infection times $t_i$, we can focus on the set of mutually *independent* random transmission times $\tau_{ji} = t_i - t_j$. Interestingly, by switching from a node-centric view to an edge-centric view, we obtain a fully factorized joint density of the set of transmission times

$$p\left(\{\tau_{ji}\}_{(j,i) \in \mathcal{E}}\right) = \prod_{(j,i) \in \mathcal{E}} f_{ji}(\tau_{ji}), \tag{2}$$

Based on the Shortest-Path property of the independent cascade model, each variable $t_i$ can be viewed as a transformation from the collection of variables $\{\tau_{ji}\}_{(j,i) \in \mathcal{E}}$.

More specifically, let $\mathcal{Q}_i$ be the collection of directed paths in $\mathcal{G}$ from the source nodes to node $i$, where each path $q \in \mathcal{Q}_i$ contains a sequence of directed edges $(j, l)$. Assuming all source nodes are infected at zero time, then we obtain variable $t_i$ via

$$t_i = g_i\left(\{\tau_{ji}\}_{(j,i) \in \mathcal{E}}\right) := \min_{q \in \mathcal{Q}_i} \sum_{(j,l) \in q} \tau_{jl}, \tag{3}$$

where the transformation $g_i(\cdot)$ is the value of the shortest-path minimization. As a special case, we can now compute the probability of node $i$ infected before $T$ using a set of independent variables:

$$\Pr\{t_i \le T\} = \Pr\left\{g_i\left(\{\tau_{ji}\}_{(j,i) \in \mathcal{E}}\right) \le T\right\}. \tag{4}$$

The significance of the relation is that it allows us to transform a problem involving a sequence of dependent variables $\{t_i\}_{i \in \mathcal{V}}$ to one with independent variables $\{\tau_{ji}\}_{(j,i) \in \mathcal{E}}$. Furthermore, the two perspectives are connected via the shortest path algorithm in weighted directed graph, a standard well-studied operation in graph analysis.

## 4   Influence Estimation Problem in Continuous-Time Diffusion Networks

Intuitively, given a time window, the wider the spread of infection, the more influential the set of sources. We adopt the definition of influence as the average number of infected nodes given a set of source nodes and a time window, as in previous work [7]. More formally, consider a set of $C$ source nodes $\mathcal{A} \subseteq \mathcal{V}$ which gets infected at time zero, then, given a time window $T$, a node $i$ is infected in the time window if $t_i \le T$. The expected number of infected nodes (or the influence) given the set of transmission functions $\{f_{ji}\}_{(j,i) \in \mathcal{E}}$ can be computed as

$$\sigma(\mathcal{A}, T) = \mathbb{E}\left[\sum_{i \in \mathcal{V}} \mathbb{I}\{t_i \le T\}\right] = \sum_{i \in \mathcal{V}} \mathbb{E}\left[\mathbb{I}\{t_i \le T\}\right] = \sum_{i \in \mathcal{V}} \Pr\{t_i \le T\}, \tag{5}$$

where $\mathbb{I}\{\cdot\}$ is the indicator function and the expectation is taken over the the set of *dependent* variables $\{t_i\}_{i \in \mathcal{V}}$.

Essentially, the influence estimation problem in Eq. (5) is an inference problem for graphical models, where the probability of event $t_i \le T$ given sources in $\mathcal{A}$ can be obtained by summing out the possible configuration of other variables $\{t_j\}_{j \ne i}$. That is

$$\Pr\{t_i \le T\} = \int_0^\infty \cdots \int_{t_i=0}^T \cdots \int_0^\infty \left(\prod_{j \in \mathcal{V}} p\left(t_j | \{t_l\}_{l \in \pi_j}\right)\right)\left(\prod_{j \in \mathcal{V}} dt_j\right), \tag{6}$$

which is, in general, a very challenging problem. First, the corresponding directed graphical models can contain nodes with high in-degree and high out-degree. For example, in Twitter, a user can follow dozens of other users, and another user can have hundreds of "followees". The tree-width corresponding to this directed graphical model can be very high, and we need to perform integration for functions involving many continuous variables. Second, the integral in general can not be eval-

uated analytically for heterogeneous transmission functions, which means that we need to resort to numerical integration by discretizing the domain $[0, \infty)$. If we use $N$ level of discretization for each variable, we would need to enumerate $O(N^{|\pi_i|})$ entries, exponential in the number of parents.

Only in very special cases, can one derive the closed-form equation for computing $\Pr\{t_i \leq T\}$ [7]. However, without further heuristic approximation, the computational complexity of the algorithm is exponential in the size and density of the network. The intrinsic complexity of the problem entails the utilization of approximation algorithms, such as mean field algorithms or message passing algorithms.We will design an efficient randomized (or sampling) algorithm in the next section.

# 5 Efficient Influence Estimation in Continuous-Time Diffusion Networks

Our first key observation is that we can transform the influence estimation problem in Eq. (5) into a problem with *independent* variables. Using relation in Eq. (4), we have

$$\sigma(\mathcal{A}, T) = \sum\nolimits_{i \in \mathcal{V}} \Pr\left\{g_i\left(\{\tau_{ji}\}_{(j,i) \in \mathcal{E}}\right) \leq T\right\} = \mathbb{E}\left[\sum\nolimits_{i \in \mathcal{V}} \mathbb{I}\left\{g_i\left(\{\tau_{ji}\}_{(j,i) \in \mathcal{E}}\right) \leq T\right\}\right], \quad (7)$$

where the expectation is with respect to the set of independent variables $\{\tau_{ji}\}_{(j,i) \in \mathcal{E}}$. This equivalent formulation suggests a naive sampling (NS) algorithm for approximating $\sigma(\mathcal{A}, T)$: draw $n$ samples of $\{\tau_{ji}\}_{(j,i) \in \mathcal{E}}$, run a shortest path algorithm for each sample, and finally average the results (see Appendix C for more details). However, this naive sampling approach has a computational complexity of $O(nC|\mathcal{V}||\mathcal{E}| + nC|\mathcal{V}|^2 \log |\mathcal{V}|)$ due to the repeated calling of the shortest path algorithm. This is quadratic to the network size, and hence not scalable to millions of nodes.

Our second key observation is that for each sample $\{\tau_{ji}\}_{(j,i) \in \mathcal{E}}$, we are only interested in the neighborhood size of the source nodes, *i.e.*, the summation $\sum_{i \in \mathcal{V}} \mathbb{I}\{\cdot\}$ in Eq. (7), rather than in the individual shortest paths. Fortunately, the neighborhood size estimation problem has been studied in the theoretical computer science literature. Here, we adapt a very efficient randomized algorithm by Cohen [17] to our influence estimation problem. This randomized algorithm has a computational complexity of $O(|\mathcal{E}| \log |\mathcal{V}| + |\mathcal{V}| \log^2 |\mathcal{V}|)$ and it estimates the neighborhood sizes for *all* possible single source node locations. Since it needs to run once for each sample of $\{\tau_{ji}\}_{(j,i) \in \mathcal{E}}$, we obtain an overall influence estimation algorithm with $O(n|\mathcal{E}| \log |\mathcal{V}| + n|\mathcal{V}| \log^2 |\mathcal{V}|)$ computation, nearly linear in network size. Next we will revisit Cohen's algorithm for neighborhood estimation.

## 5.1 Randomized Algorithm for Single-Source Neighborhood Size Estimation

Given a fixed set of edge transmission times $\{\tau_{ji}\}_{(j,i) \in \mathcal{E}}$ and a source node $s$, infected at time 0, the neighborhood $\mathcal{N}(s, T)$ of a source node $s$ given a time window $T$ is the set of nodes within distance $T$ from $s$, *i.e.*,

$$\mathcal{N}(s, T) = \left\{i \mid g_i\left(\{\tau_{ji}\}_{(j,i) \in \mathcal{E}}\right) \leq T, \; i \in \mathcal{V}\right\}. \quad (8)$$

Instead of estimating $\mathcal{N}(s, T)$ directly, the algorithm will assign an exponentially distributed random label $r_i$ to each network node $i$. Then, it makes use of the fact that the minimum of a set of exponential random variables $\{r_i\}_{i \in \mathcal{N}(s,T)}$ will also be a exponential random variable, but with its parameter equals to the number of variables. That is if each $r_i \sim \exp(-r_i)$, then the smallest label within distance $T$ from source $s$, $r_* := \min_{i \in \mathcal{N}(s,T)} r_i$, will distribute as $r_* \sim \exp\{-|\mathcal{N}(s, T)| r_*\}$. Suppose we randomize over the labeling $m$ times, and obtain $m$ such least labels, $\{r_*^u\}_{u=1}^m$. Then the neighborhood size can be estimated as

$$|\mathcal{N}(s, T)| \approx \frac{m - 1}{\sum_{u=1}^m r_*^u}. \quad (9)$$

which is shown to be an unbiased estimator of $|\mathcal{N}(s, T)|$ [17]. This is an interesting relation since it allows us to transform the counting problem in (8) to a problem of finding the minimum random label $r_*$. The key question is whether we can compute the least label $r_*$ efficiently, given random labels $\{r_i\}_{i \in \mathcal{V}}$ and any source node $s$.

Cohen [17] designed a modified Dijkstra's algorithm (Algorithm 1) to construct a data structure $r_*(s)$, called least label list, for each node $s$ to support such query. Essentially, the algorithm starts with the node $i$ with the smallest label $r_i$, and then it traverses in breadth-first search fashion along the reverse direction of the graph edges to find all reachable nodes. For each reachable node $s$, the distance $d_*$ between $i$ and $s$, and $r_i$ are added to the end of $r_*(s)$. Then the algorithm moves to the node $i'$ with the second smallest label $r_{i'}$, and similarly find all reachable nodes. For each reachable

node $s$, the algorithm will compare the current distance $d_*$ between $i'$ and $s$ with the last recorded distance in $r_*(s)$. If the current distance is smaller, then the current $d_*$ and $r_{i'}$ are added to the end of $r_*(s)$. Then the algorithm move to the node with the third smallest label and so on. The algorithm is summarized in Algorithm 1 in Appendix D.

Algorithm 1 returns a list $r_*(s)$ per node $s \in \mathcal{V}$, which contains information about distance to the smallest reachable labels from $s$. In particular, each list contains pairs of distance and random labels, $(d, r)$, and these pairs are ordered as

$$\infty > d_{(1)} > d_{(2)} > \ldots > d_{(|r_*(s)|)} = 0 \tag{10}$$

$$r_{(1)} < r_{(2)} < \ldots < r_{(|r_*(s)|)}, \tag{11}$$

where $\{\cdot\}_{(l)}$ denotes the $l$-th element in the list. (see Appendix D for an example). If we want to query the smallest reachable random label $r_*$ for a given source $s$ and a time $T$, we only need to perform a binary search on the list for node $s$:

$$r_* = r_{(l)}, \text{ where } d_{(l-1)} > T \geq d_{(l)}. \tag{12}$$

Finally, to estimate $|\mathcal{N}(s, T)|$, we generate $m$ *i.i.d.* collections of random labels, run Algorithm 1 on each collection, and obtain $m$ values $\{r_*^u\}_{u=1}^m$, which we use on Eq. (9) to estimate $|\mathcal{N}(i, T)|$.

The computational complexity of Algorithm 1 is $O(|\mathcal{E}| \log |\mathcal{V}| + |\mathcal{V}| \log^2 |\mathcal{V}|)$, with expected size of each $r_*(s)$ being $O(\log |\mathcal{V}|)$. Then the expected time for querying $r_*$ is $O(\log \log |\mathcal{V}|)$ using binary search. Since we need to generate $m$ set of random labels and run Algorithm 1 $m$ times, the overall computational complexity for estimating the single-source neighborhood size for all $s \in \mathcal{V}$ is $O(m|\mathcal{E}| \log |\mathcal{V}| + m|\mathcal{V}| \log^2 |\mathcal{V}| + m|\mathcal{V}| \log \log |\mathcal{V}|)$. For large scale network, and when $m \ll \min\{|\mathcal{V}|, |\mathcal{E}|\}$, this randomized algorithm can be much more efficient than approaches based on directly calculating the shortest paths.

## 5.2 Constructing Estimation for Multiple-Source Neighborhood Size

When we have a set of sources, $\mathcal{A}$, its neighborhood is the union of the neighborhoods of its constituent sources

$$\mathcal{N}(\mathcal{A}, T) = \bigcup\nolimits_{i \in \mathcal{A}} \mathcal{N}(i, T). \tag{13}$$

This is true because each source independently infects its downstream nodes. Furthermore, to calculate the least label list $r_*$ corresponding to $\mathcal{N}(\mathcal{A}, T)$, we can simply reuse the least label list $r_*(i)$ of each individual source $i \in \mathcal{A}$. More formally,

$$r_* = \min_{i \in \mathcal{A}} \ \min_{j \in \mathcal{N}(i, T)} r_j, \tag{14}$$

where the inner minimization can be carried out by querying $r_*(i)$. Similarly, after we obtain $m$ samples of $r_*$, we can estimate $|\mathcal{N}(\mathcal{A}, T)|$ using Eq. (9). Importantly, very little additional work is needed when we want to calculate $r_*$ for a set of sources $\mathcal{A}$, and we can reuse work done for a single source. This is very different from a naive sampling approach where the sampling process needs to be done completely anew if we increase the source set. In contrast, using the randomized algorithm, only an additional constant-time minimization over $|\mathcal{A}|$ numbers is needed.

## 5.3 Overall Algorithm

So far, we have achieved efficient neighborhood size estimation of $|\mathcal{N}(\mathcal{A}, T)|$ with respect to a given set of transmission times $\{\tau_{ji}\}_{(j,i) \in \mathcal{E}}$. Next, we will estimate the influence by averaging over multiple sets of samples for $\{\tau_{ji}\}_{(j,i) \in \mathcal{E}}$. More specifically, the relation from (7)

$$\sigma(\mathcal{A}, T) = \mathbb{E}_{\{\tau_{ji}\}_{(j,i) \in \mathcal{E}}} \left[ |\mathcal{N}(\mathcal{A}, T)| \right] = \mathbb{E}_{\{\tau_{ji}\}} \mathbb{E}_{\{r^1, \ldots, r^m\} | \{\tau_{ji}\}} \left[ \frac{m-1}{\sum_{u=1}^m r_*^u} \right], \tag{15}$$

suggests the following overall algorithm

---

Continuous-Time Influence Estimation (CONTINEST):

 1. Sample $n$ sets of random transmission times $\{\tau_{ij}^l\}_{(j,i) \in \mathcal{E}} \ \sim \ \prod_{(j,i) \in \mathcal{E}} f_{ji}(\tau_{ji})$
 2. Given a set of $\{\tau_{ij}^l\}_{(j,i) \in \mathcal{E}}$, sample $m$ sets of random labels $\{r_i^u\}_{i \in \mathcal{V}} \ \sim \ \prod_{i \in \mathcal{V}} \exp(-r_i)$
 3. Estimate $\sigma(\mathcal{A}, T)$ by sample averages $\sigma(\mathcal{A}, T) \approx \frac{1}{n} \sum_{l=1}^n \left( (m-1) / \sum_{u_l=1}^m r_*^{u_l} \right)$

---

Importantly, the number of random labels, $m$, does not need to be very large. Since the estimator for $|\mathcal{N}(A, T)|$ is unbiased [17], essentially the outer-loop of averaging over $n$ samples of random transmission times further reduces the variance of the estimator in a rate of $O(1/n)$. In practice, we can use a very small $m$ (*e.g.*, 5 or 10) and still achieve good results, which is also confirmed by our later experiments. Compared to [2], the novel application of Cohen's algorithm arises for estimating influence for multiple sources, which drastically reduces the computation by cleverly using the least-label list from single source. Moreover, we have the following theoretical guarantee (see Appendix E for proof)

**Theorem 1** *Draw the following number of samples for the set of random transmission times*

$$n \geqslant \frac{C\Lambda}{\epsilon^2} \log\left(\frac{2|\mathcal{V}|}{\delta}\right) \tag{16}$$

*where $\Lambda := \max_{\mathcal{A}:|\mathcal{A}| \leq C} 2\sigma(\mathcal{A}, T)^2/(m-2) + 2Var(|\mathcal{N}(\mathcal{A}, T)|)(m-1)/(m-2) + 2a\epsilon/3$ and $|\mathcal{N}(\mathcal{A}, T)| \leq a$, and for each set of random transmission times, draw $m$ set of random labels. Then $|\widehat{\sigma}(\mathcal{A}, T) - \sigma(\mathcal{A}, T)| \leqslant \epsilon$ uniformly for all $\mathcal{A}$ with $|\mathcal{A}| \leqslant C$, with probability at least $1 - \delta$.*

The theorem indicates that the minimum number of samples, $n$, needed to achieve certain accuracy is related to the actual size of the influence $\sigma(\mathcal{A}, T)$, and the variance of the neighborhood size $|\mathcal{N}(\mathcal{A}, T)|$ over the random draw of samples. The number of random labels, $m$, drawn in the inner loop of the algorithm will monotonically decrease the dependency of $n$ on $\sigma(\mathcal{A}, T)$. It suffices to draw a small number of random labels, as long as the value of $\sigma(\mathcal{A}, T)^2/(m-2)$ matches that of $Var(|\mathcal{N}(\mathcal{A}, T)|)$. Another implication is that influence at larger time window $T$ is harder to estimate, since $\sigma(\mathcal{A}, T)$ will generally be larger and hence require more random labels.

## 6 Influence Maximization

Once we know how to estimate the influence $\sigma(\mathcal{A}, T)$ for any $\mathcal{A} \subseteq \mathcal{V}$ and time window $T$ efficiently, we can use them in finding the optimal set of $C$ source nodes $\mathcal{A}^* \subseteq \mathcal{V}$ such that the expected number of infected nodes in $\mathcal{G}$ is maximized at $T$. That is, we seek to solve,

$$\mathcal{A}^* = \mathrm{argmax}_{|\mathcal{A}| \leqslant C} \ \sigma(\mathcal{A}, T), \tag{17}$$

where set $\mathcal{A}$ is the variable. The above optimization problem is NP-hard in general. By construction, $\sigma(\mathcal{A}, T)$ is a non-negative, monotonic nondecreasing function in the set of source nodes, and it can be shown that $\sigma(\mathcal{A}, T)$ satisfies a diminishing returns property called submodularity [7].

A well-known approximation algorithm to maximize monotonic submodular functions is the *greedy algorithm*. It adds nodes to the source node set $\mathcal{A}$ sequentially. In step $k$, it adds the node $i$ which maximizes the *marginal gain* $\sigma(\mathcal{A}_{k-1} \cup \{i\}; T) - \sigma(\mathcal{A}_{k-1}; T)$. The greedy algorithm finds a source node set which achieves at least a constant fraction $(1 - 1/e)$ of the optimal [18]. Moreover, lazy evaluation [5] can be employed to reduce the required number of *marginal gains* per iteration. By using our influence estimation algorithm in each iteration of the greedy algorithm, we gain the following additional benefits:

First, at each iteration $k$, we do not need to rerun the full influence estimation algorithm (section 5.2). We just need to store the least label list $r_*(i)$ for each node $i \in \mathcal{V}$ computed for a single source, which requires expected storage size of $O(|\mathcal{V}| \log |\mathcal{V}|)$ overall.

Second, our influence estimation algorithm can be easily parallelized. Its two nested sampling loops can be parallelized in a straightforward way since the variables are independent of each other. However, in practice, we use a small number of random labels, and $m \ll n$. Thus we only need to parallelize the sampling for the set of random transmission times $\{\tau_{ji}\}$. The storage of the least element lists can also be distributed.

However, by using our randomized algorithm for influence estimation, we also introduce a sampling error to the greedy algorithm due to the approximation of the influence $\sigma(\mathcal{A}, T)$. Fortunately, the greedy algorithm is tolerant to such sampling noise, and a well-known result provides a guarantee for this case (following an argument in [19, Th. 7.9]):

**Theorem 2** *Suppose the influence $\sigma(\mathcal{A}, T)$ for all $\mathcal{A}$ with $|\mathcal{A}| \leq C$ are estimated uniformly with error $\epsilon$ and confidence $1 - \delta$, the greedy algorithm returns a set of sources $\widehat{\mathcal{A}}$ such that $\sigma(\widehat{\mathcal{A}}, T) \geq (1 - 1/e)OPT - 2C\epsilon$ with probability at least $1 - \delta$.*

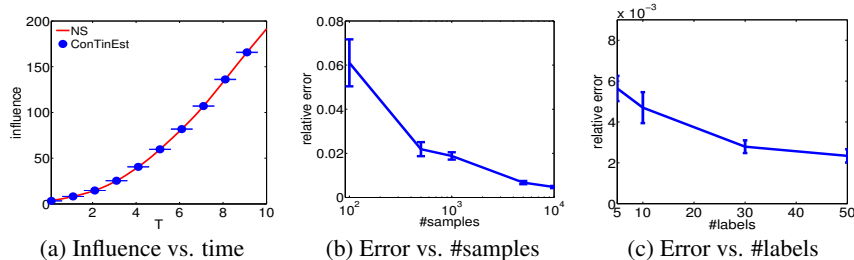

| (a) Influence vs. time | (b) Error vs. #samples | (c) Error vs. #labels |

Figure 1: For core-periphery networks with 1,024 nodes and 2,048 edges, (a) estimated influence for increasing time window $T$, and (b) fixing $T = 10$, relative error for increasing number of samples with 5 random labels, and (c) for increasing number of random labels with 10,000 random samples.

## 7 Experiments

We evaluate the accuracy of the estimated influence given by CONTINEST and investigate the performance of influence maximization on synthetic and real networks. We show that our approach significantly outperforms the state-of-the-art methods in terms of both speed and solution quality.

**Synthetic network generation.** We generate three types of Kronecker networks [20]: (*i*) core-periphery networks (parameter matrix: [0.9 0.5; 0.5 0.3]), which mimic the information diffusion traces in real world networks [21], (*ii*) random networks ([0.5 0.5; 0.5 0.5]), typically used in physics and graph theory [22] and (*iii*) hierarchical networks ([0.9 0.1; 0.1 0.9]) [10]. Next, we assign a pairwise transmission function for every directed edge in each type of network and set its parameters at random. In our experiments, we use the Weibull distribution [16], $f(t; \alpha, \beta) = \frac{\beta}{\alpha} \left( \frac{t}{\alpha} \right)^{\beta-1} e^{-(t/\alpha)^\beta}, t \geqslant 0$, where $\alpha > 0$ is a scale parameter and $\beta > 0$ is a shape parameter. The Weibull distribution (Wbl) has often been used to model lifetime events in survival analysis, providing more flexibility than an exponential distribution [16]. We choose $\alpha$ and $\beta$ from 0 to 10 uniformly at random for each edge in order to have heterogeneous temporal dynamics. Finally, for each type of Kronecker network, we generate 10 sample networks, each of which has different $\alpha$ and $\beta$ chosen for every edge.

**Accuracy of the estimated influence.** To the best of our knowledge, there is no analytical solution to the influence estimation given Weibull transmission function. Therefore, we compare CONTINEST with Naive Sampling (NS) approach (see Appendix C) by considering the highest degree node in a network as the source, and draw 1,000,000 samples for NS to obtain near ground truth. Figures 1(a) compares CONTINEST with the ground truth provided by NS at different time window $T$, from 0.1 to 10 in corre-periphery networks. For CONTINEST, we generate up to 10,000 random samples (or set of random waiting times), and 5 random labels in the inner loop. In all three networks, estimation provided by CONTINEST fits accurately the ground truth, and the relative error decreases quickly as we increase the number of samples and labels (Figures 1(b) and 1(c)). For 10,000 random samples with 5 random labels, the relative error is smaller than 0.01. (see Appendix F for additional results on the random and hierarchal networks)

**Scalability.** We compare CONTINEST to the state-of-the-art method INFLUMAX [7] and the Naive Sampling (NS) method in terms of runtime for the continuous-time influence estimation and maximization. For CONTINEST, we draw 10,000 samples in the outer loop, each having 5 random labels in the inner loop. For NS, we also draw 10,000 samples. The first two experiments are carried out in a single 2.4GHz processor. First, we compare the performance of increasingly selecting sources (from 1 to 10) on small core-periphery networks (Figure 2(a)). When the number of selected sources is 1, different algorithms essentially spend time estimating the influence for each node. CONTINEST outperforms other methods by order of magnitude and for the number of sources larger than 1, it can efficiently reuse computations for estimating influence for individual nodes. Dashed lines mean that a method did not finish in 24 hours, and the estimated run time is plotted. Next, we compare the run time for selecting 10 sources on core-periphery networks of 128 nodes with increasing densities (or the number of edges) (Figure 2(a)). Again, INFLUMAX and NS are order of magnitude slower due to their respective exponential and quadratic computational complexity in network density. In contrast, the run time of CONTINEST only increases slightly with the increasing density since its computational complexity is linear in the number of edges (see Appendix F for additional results on the random and hierarchal networks). Finally, we evaluate the speed on large core-periphery networks, ranging from 100 to 1,000,000 nodes with density 1.5 in Figure 2(c). We report the parallel run time

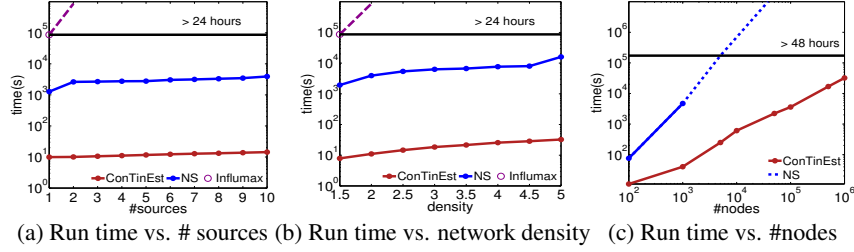

(a) Run time vs. # sources  (b) Run time vs. network density  (c) Run time vs. #nodes

Figure 2: For core-periphery networks with $T = 10$, runtime for (a) selecting increasing number of sources in networks of 128 nodes and 320 edges; for (b)selecting 10 sources in networks of 128 nodes with increasing density; and for (c) selecting 10 sources with increasing network size from 100 to 1,000,000 fixing 1.5 density.

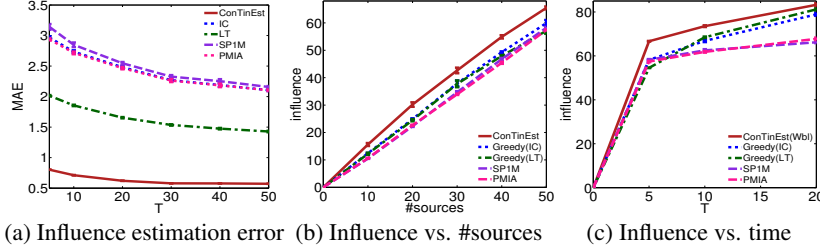

(a) Influence estimation error  (b) Influence vs. #sources  (c) Influence vs. time

Figure 3: In MemeTracker dataset, (a) comparison of the accuracy of the estimated influence in terms of mean absolute error, (b) comparison of the influence of the selected nodes by fixing the observation window $T = 5$ and varying the number sources, and (c) comparison of the influence of the selected nodes by by fixing the number of sources to 50 and varying the time window.

only for CONTINEST and NS (both are implemented by MPI running on 192 cores of 2.4Ghz) since INFLUMAX is not scalable. In contrast to NS, the performance of CONTINEST increases linearly with the network size and can easily scale up to one million nodes.

**Real-world data.** We first quantify how well each method can estimate the true influence in a real-world dataset. Then, we evaluate the solution quality of the selected sources for influence maximization. We use the MemeTracker dataset [23] which has 10,967 hyperlink cascades among 600 media sites. We repeatedly split all cascades into a 80% training set and a 20% test set at random for five times. On each training set, we learn the continuous-time model using NETRATE [10] with exponential transmission functions. For discrete-time model, we learn the infection probabilities using [24] for IC, SP1M and PMIA. Similarly, for LT, we follow the methodology by [1]. Let $\mathcal{C}(u)$ be the set of all cascades where $u$ was the source node. Based on $\mathcal{C}(u)$, the total number of distinct nodes infected before $T$ quantifies the real influence of node $u$ up to time $T$. In Figure 3(a), we report the Mean Absolute Error (MAE) between the real and the estimated influence. Clearly, CONTINEST performs the best statistically. Because the length of real cascades empirically conforms to a power-law distribution where most cascades are very short (2-4 nodes), the gap of the estimation error is relatively not large. However, we emphasize that such accuracy improvement is critical for maximizing long-term influence. The estimation error for individuals will accumulate along the spreading paths. Hence, any consistent improvement in influence estimation can lead to significant improvement to the overall influence estimation and maximization task, which is further confirmed by Figures 3(b) and 3(c) where we evaluate the influence of the selected nodes in the same spirit as influence estimation: the true influence is calculated as the total number of distinct nodes infected before $T$ based on $\mathcal{C}(u)$ of the selected nodes. The selected sources given by CONTINEST achieve the best performance as we vary the number of selected sources and the observation time window.

## 8  Conclusions

We propose a randomized influence estimation algorithm in continuous-time diffusion networks, which can scale up to networks of millions of nodes while significantly improves over previous state-of-the-arts in terms of the accuracy of the estimated influence and the quality of the selected nodes in maximizing the influence. In future work, it will be interesting to apply the current algorithm to other tasks like influence minimization and manipulation, and design scalable algorithms for continuous-time models other than the independent cascade model.

**Acknowledgement:** Our work is supported by NSF/NIH BIGDATA 1R01GM108341-01, NSF IIS1116886, NSF IIS1218749, NSFC 61129001, a DARPA Xdata grant and Raytheon Faculty Fellowship of Gatech.

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
