[Supplementary Material]

# A    Heterogeneous Transmission Functions

We denote the waiting time distribution, or transmission function, along a directed edge of $\mathcal{G}$ as $f_{ji}(t_i|t_j)$. Formally, the transmission function $f_{ji}(t_i|t_j)$ for directed edge $j \to i$ is the conditional density of node $i$ getting infected at time $t_i$ given that node $j$ was infected at time $t_j$. We assume it is shift invariant, *i.e.*, $f_{ji}(t_i|t_j) = f_{ji}(t_i - t_j) = f_{ji}(\tau_{ji})$, where $\tau_{ji} := t_i - t_j$, and it takes positive values when $\tau_{ji} \geq 0$, and the value of zero otherwise.

In most previous work, simple parametric transmission functions such as the exponential distribution $\alpha_{ji} \exp(-\alpha_{ji}\tau_{ji})$, and the Rayleigh distribution $\alpha_{ji}\tau \exp(-\alpha_{ji}\tau_{ji}^2/2)$ have been used [16, 10]. However, in many real world scenarios, information transmission between pairs of nodes can be heterogeneous and the waiting times can obey distributions that dramatically differ from these simple models. For instance, in viral marketing, active consumers could update their status instantly, while an inactive user may just log in and respond once a day. As a result, the transmission function between an active user and his friends can be quite different from that between an inactive user and his friends. As an attempt to model these complex scenarios, nonparametric transmission functions have been recently considered [8]. In such approach, the relationship between the survival function, the conditional intensity function or hazard, and the transmission function is exploited. In particular, the survival function is defined as $S_{ji}(\tau_{ji}) := 1 - \int_0^{\tau_{ji}} f_{ji}(\tau')d\tau'$ and the hazard function is defined as $h_{ji}(\tau_{ji}) := f_{ji}(\tau_{ji})/S_{ji}(\tau_{ji})$. Then, it is a well-known result in survival theory that $S_{ji}(\tau_{ji}) = \exp\left(-\int_0^{\tau_{ji}} h_{ji}(\tau')d\tau'\right)$ and $f_{ji}(\tau_{ji}) = h_{ji}(\tau_{ji})S_{ji}(\tau_{ji})$. The advantage of using the conditional intensity function is that we do not need to explicitly enforce "the integral equals 1" constraint for the conditional density $f_{ji}$. Instead, we just need to ensure $h_{ji} \geq 0$. This facilitates nonparametric modeling of the transmission function. For instance, we can define the conditional intensity function as a positive combination of $n$ positive kernel functions $k$,

$$h_{ji}(\tau) = \sum\nolimits_{l=1}^{n} \alpha_l k(\tau_l, \tau), \text{ if } \tau > 0, \text{ and } 0 \text{ otherwise.}$$

A common choice of the kernel function is the Gaussian RBF kernel $k(\tau', \tau) = \exp(-\|\tau - \tau'\|^2/2s^2)$. Nonparametric transmission functions significantly improve modeling of real world diffusion, as is shown in [8].

# B    A Graphical Model Perspective

Now, we look at the independent cascade model from the perspective of graphical models, where the collection of random variables includes the infection times $t_i$ of the nodes. Although the original contact graph $\mathcal{G}$ can contain directed loops, each diffusion process (or a cascade) induces a directed acyclic graph (DAG). For those cascades consistent with a particular DAG, we can model the joint density of $t_i$ using a directed graphical model:

$$p\left(\{t_i\}_{i \in \mathcal{V}}\right) = \prod\nolimits_{i \in \mathcal{V}} p\left(t_i|\{t_j\}_{j \in \pi_i}\right), \tag{18}$$

where each $\pi_i$ denotes the collection of parents of node $i$ in the induced DAG, and each term $p(t_i|\{t_j\}_{j \in \pi_i})$ corresponds to a conditional density of $t_j$ given the infection times of the parents of node $i$. This is true because given the infection times of node $i$'s parents, $t_i$ is independent of other infection times, satisfying the local Markov property of a directed graphical model. We note that the independent cascade model only specifies explicitly the pairwise transmission functions for each directed edge, but does not directly define the conditional density $p(t_i|\{t_j\}_{j \in \pi_i})$.

However, these conditional densities can be derived from the pairwise transmission functions based on the Independent-Infection property [10]:

$$p\left(t_i|\{t_j\}_{j \in \pi_i}\right) = \sum\nolimits_{j \in \pi_i} h_{ji}(t_i|t_j) \prod\nolimits_{l \in \pi_i} S(t_i|t_l), \tag{19}$$

which is the sum of the likelihoods that node $i$ is infected by each parent node $j$. More precisely, each term in the summation can be interpreted as the instantaneous risk of node $i$ being infected at $t_i$ by node $j$ given that it has survived the infection of all parent nodes until time $t_i$.

Perhaps surprisingly, the factorization in Eq. (18) is the same factorization that can be used for an arbitrary induced DAG consistent with the contact network $\mathcal{G}$. In this case, we only need to replace the definition of $\pi_i$ (the parent of node $i$ in the DAG) to the set of neighbors of node $i$ with an edge

pointing to node $i$ in $\mathcal{G}$. This is not immediately obvious from Eq. (18), since the contact network $\mathcal{G}$ can contain directed loops which may be in conflict with the conditional independence semantics of directed graphical models. The reason it is possible to do so is as follows: Any fixed set of infection times, $t_1, \ldots, t_d$, induces an ordering of the infection times. If $t_i \leq t_j$ for an edge $j \rightarrow i$ in $\mathcal{G}$, $h_{ji}(t_i|t_j) = 0$, and the corresponding term in Eq. (19) is zeroed out, making the conditional density consistent with the semantics of directed graphical models.

Based on the joint density of the infection times in Eq. (18), we can perform various inference and learning tasks. For instance, previous work has used Eq. (18) for learning the parameters of the independent cascade model [8, 10, 11]. However, this may not be the most convenient form for addressing other inference problems, including the influence estimation problem in the next section. To this end, we propose an alternative view.

Instead of directly modeling the infection times $t_i$, we can focus on the collection of mutually independent random transmission times $\tau_{ji} = t_i - t_j$. In this case, the joint density of the collection of transmission times $\tau_{ji}$ is fully factorized

$$p\left(\{\tau_{ji}\}_{(j,i)\in\mathcal{E}}\right) = \prod\nolimits_{(j,i)\in\mathcal{E}} f_{ji}(\tau_{ji}),$$

where $\mathcal{E}$ denotes the set of edges in the contact network $\mathcal{G}$ — switching from the earlier node-centric view to the now edge-centric view. Based on the Shortest-Path property of the independent cascade model, variable $t_i$ can be viewed as a transformation from the collection of variables $\{\tau_{ji}\}_{(j,i)\in\mathcal{E}}$. More specifically, let $\mathcal{Q}_i$ be the collection of directed paths in $\mathcal{G}$ from the source nodes to node $i$, where each path $q \in \mathcal{Q}_i$ contains a sequence of directed edges $(j, l)$, and assuming all source nodes are infected at zero time, then we obtain variable $t_i$ via

$$t_i = g_i\left(\{\tau_{ji}\}_{(j,i)\in\mathcal{E}}\right) := \min_{q\in\mathcal{Q}_i} \sum\nolimits_{(j,l)\in q} \tau_{jl}, \tag{20}$$

where $g_i(\cdot)$ is the transformation.

Importantly, we can now compute the probability of infection of node $i$ at $t_i$ using the set of variables $\{\tau_{ji}\}_{(j,i)\in\mathcal{E}}$:

$$\Pr\{t_i \leq T\} = \Pr\left\{g_i\left(\{\tau_{ji}\}_{(j,i)\in\mathcal{E}}\right) \leq T\right\}. \tag{21}$$

The significance of the relation is that it allows us to transform a problem involving a sequence of dependent variables $\{t_i\}_{i\in\mathcal{V}}$ to one with independent variables $\{\tau_{ji}\}_{(j,i)\in\mathcal{E}}$. Furthermore, the two problems are connected via the shortest path algorithm in weighted directed graph, a standard well studied operation in graph analysis.

## C   Naive Sampling Algorithm

The graphical model perspective described in Section 3 and Appendix B suggests a naive sampling (NS) algorithm for approximating $\sigma(\mathcal{A}, T)$:

1. Draw $n$ samples, $\left\{\{\tau_{ji}^l\}_{(j,i)\in\mathcal{E}}\right\}_{l=1}^n$, *i.i.d.* from the waiting time product distribution $\prod_{(j,i)\in\mathcal{E}} f_{ji}(\tau_{ji})$;

2. For each sample $\{\tau_{ji}^l\}_{(j,i)\in\mathcal{E}}$ and for each node $i$, find the shortest path from source nodes to node $i$; count the number of nodes with $g_i\left(\{\tau_{ji}^l\}_{(j,i)\in\mathcal{E}}\right) \leq T$;

3. Average the counts across $n$ samples.

Although the naive sampling algorithm can handle arbitrary transmission function, it is not scalable to networks with millions of nodes. We need to compute the shortest path for each node and each sample, which results in a computational complexity of $O(n|\mathcal{E}| + n|\mathcal{V}|\log|\mathcal{V}|)$ for a single source node. The problem is even more pressing in the influence maximization problem, where we need to estimate the influence of source nodes at different location and with increasing number of source nodes. To do this, the algorithm needs to be repeated, adding a multiplicative factor of $C|\mathcal{V}|$ to the computational complexity ($C$ is the number of nodes to select). Then, the algorithm becomes quadratic in the network size. When the network size is in the order of thousands and millions, typical in modern social network analysis, the naive sampling algorithm become prohibitively ex-

- Node labeling :
  $e(0.2) < b(0.3) < d(0.4) < a(1.5) < c(1.8) < g(2.2) < f(3.7)$

- Neighborhoods:
  $\mathcal{N}(c,2) = \{a,b,c,e\}$; $\mathcal{N}(c,3) = \{a,b,c,d,e,f\}$;

- Least-label list:
  $r_*(c) : (2,0.2), (1,0.3), (0.5,1.5), (0,1.8)$

- Query: $r_*(c,0.8) = r(a) = 1.5$

Figure 4: Graph $\mathcal{G} = (\mathcal{V}, \mathcal{E})$, edge weights $\{\tau_{ji}\}_{(j,i)\in\mathcal{E}}$, and node labeling $\{r_i\}_{i\in\mathcal{V}}$ with the associated output from Algorithm 1.

pensive. Additionally, we may need to draw thousands of samples ($n$ is large), further making the algorithm impractical for large scale problems.

## D  Least Label List

The notation "argsort$((r_1,\ldots,r_{|\mathcal{V}|}),\text{ascend})$" in line 2 of Algorithm 1 means that we sort the collection of random labels in ascending order and return the argument of the sort as an ordered list.

---

**Algorithm 1:** Least Label List

---

**Input**: a reversed directed graph $\mathcal{G} = (\mathcal{V}, \mathcal{E})$ with edge weights $\{\tau_{ji}\}_{(j,i)\in\mathcal{E}}$, a node labeling
$\qquad \{r_i\}_{i\in\mathcal{V}}$
**Output**: A list $r_*(s)$ for each $s \in \mathcal{V}$
**for** *each $s \in \mathcal{V}$* **do** $d_s \leftarrow \infty, r_*(s) \leftarrow \emptyset$
**for** *$i$ in argsort$((r_1,\ldots,r_{|\mathcal{V}|}),\text{ascend})$* **do**
$\quad$ empty heap $\mathtt{H} \leftarrow \emptyset$;
$\quad$ set all nodes except $i$ as unvisited;
$\quad$ push $(0,i)$ into heap $\mathtt{H}$;
$\quad$ **while** $\mathtt{H} \neq \emptyset$ **do**
$\quad\quad$ pop $(d_*, s)$ with the minimum $d_*$ from $\mathtt{H}$;
$\quad\quad$ add $(d_*, r_i)$ to the end of list $r_*(s)$;
$\quad\quad$ $d_s \leftarrow d^*$;
$\quad\quad$ **for** *each unvisited out-neighbor $j$ of $s$* **do**
$\quad\quad\quad$ set $j$ as visited;
$\quad\quad\quad$ **if** *$(d,j)$ in heap $\mathtt{H}$* **then**
$\quad\quad\quad\quad$ Pop $(d,j)$ from heap $\mathtt{H}$;
$\quad\quad\quad\quad$ Push $(\min\{d, d_* + \tau_{js}\}, j)$ into heap $\mathtt{H}$;
$\quad\quad\quad$ **else if** $d_* + \tau_{js} < d_j$ **then**
$\quad\quad\quad\quad$ Push $(d_* + \tau_{js}, j)$ into heap $\mathtt{H}$;

---

Figure 4 shows an example of the Least-Label-List. The nodes from $a$ to $g$ are assigned to exponentially distributed labels with mean one shown in each parentheses. Given a query distance 0.8 for node $c$, we can binary-search its Least-label-list $r_*(c)$ to find that node $a$ belongs to this range with the smallest label $r(a) = 1.5$.

# E  Theorem 1

**Theorem 1** *Sample the following number of sets of random transmission times*

$$n \geqslant \frac{C\Lambda}{\epsilon^2} \log\left(\frac{2|\mathcal{V}|}{\delta}\right)$$

*where* $\Lambda := \max_{\mathcal{A}:|\mathcal{A}|\leq C} 2\sigma(\mathcal{A},T)^2/(m-2) + 2Var(|\mathcal{N}(\mathcal{A},T)|)(m-1)/(m-2) + 2a\epsilon/3$, $|\mathcal{N}(\mathcal{A},T)| \leqslant a$, *and for each set of random transmission times, sample* $m$ *set of random labels. Then we can guarantee that*

$$|\widehat{\sigma}(\mathcal{A},T) - \sigma(\mathcal{A},T)| \leqslant \epsilon$$

*simultaneously for all* $\mathcal{A}$ *with* $|\mathcal{A}| \leqslant C$, *with probability at least* $1 - \delta$.

**Proof** Let $S_\tau := |\mathcal{N}(\mathcal{A},T)|$ for a fixed set of $\{\tau_{ji}\}$ and then $\sigma(\mathcal{A},T) = \mathbb{E}_\tau[S_\tau]$. The randomized algorithm with $m$ randomizations produces an unbiased estimator $\widehat{S}_\tau = (m-1)/(\sum_{u=1}^m r_*^u)$ for $S_\tau$, *i.e.*, $\mathbb{E}_{r|\tau}[\widehat{S}_\tau] = S_\tau$, with variance $\mathbb{E}_{r|\tau}[(\widehat{S}_\tau - S_\tau)^2] = S_\tau^2/(m-2)$.

Then $\widehat{S}_\tau$ is also an unbiased estimator for $\sigma(\mathcal{A},T)$, since $\mathbb{E}_{\tau,r}[\widehat{S}_\tau] = \mathbb{E}_\tau\mathbb{E}_{r|\tau}[\widehat{S}_\tau] = \mathbb{E}_\tau[S_\tau] = \sigma(\mathcal{A},T)$. Its variance is

$$\begin{aligned}
Var(\widehat{S}_\tau) &:= \mathbb{E}_{\tau,r}[(\widehat{S}_\tau - \sigma(\mathcal{A},T))^2] = \mathbb{E}_{\tau,r}[(\widehat{S}_\tau - S_\tau + S_\tau - \sigma(\mathcal{A},T))^2] \\
&= \mathbb{E}_{\tau,r}[(\widehat{S}_\tau - S_\tau)^2] + 2\,\mathbb{E}_{\tau,r}[(\widehat{S}_\tau - S_\tau)(S_\tau - \sigma(\mathcal{A},T))] + \mathbb{E}_{\tau,r}[(S_\tau - \sigma(\mathcal{A},T))^2] \\
&= \mathbb{E}_\tau[S_\tau^2/(m-2)] + 0 + Var(S_\tau) \\
&= \sigma(\mathcal{A},T)^2/(m-2) + Var(S_\tau)(m-1)/(m-2)
\end{aligned}$$

Then using Bernstein's inequality, we have, for our final estimator $\widehat{\sigma}(\mathcal{A},T) = \frac{1}{n}\sum_{l=1}^n \widehat{S}_{\tau^l}$, that

$$\Pr\left\{|\widehat{\sigma}(\mathcal{A},T) - \sigma(\mathcal{A},T)| \geqslant \epsilon\right\} \leqslant 2\exp\left(-\frac{n\epsilon^2}{2Var(\widehat{S}_\tau) + 2a\epsilon/3}\right) \tag{22}$$

where $\widehat{S}_\tau < a \leqslant |\mathcal{V}|$.

Setting the right hand side of relation (22) to $\delta$, we have that, with probability $1 - \delta$, sampling the following number sets of random transmission times

$$n \geqslant \frac{2Var(\widehat{S}_\tau) + 2a\epsilon/3}{\epsilon^2}\log\left(\frac{2}{\delta}\right) = \frac{2\sigma(\mathcal{A},T)^2/(m-2) + 2Var(S_\tau)(m-1)/(m-2) + 2a\epsilon/3}{\epsilon^2}\log\left(\frac{2}{\delta}\right)$$

we can guarantee that our estimator to have error $|\widehat{\sigma}(\mathcal{A},T) - \sigma(\mathcal{A},T)| \leqslant \epsilon$.

If we want to insure that $|\widehat{\sigma}(\mathcal{A},T) - \sigma(\mathcal{A},T)| \leqslant \epsilon$ simultaneously hold for all $\mathcal{A}$ such that $|\mathcal{A}| \leqslant C \ll |\mathcal{V}|$, we can first use union bound with relation (22). In this case, we have that, with probability $1 - \delta$, sampling the following number sets of random transmission times

$$n \geqslant \frac{C\Lambda}{\epsilon^2} \log\left(\frac{2|\mathcal{V}|}{\delta}\right)$$

we can guarantee that our estimator to have error $|\widehat{\sigma}(\mathcal{A},T) - \sigma(\mathcal{A},T)| \leqslant \epsilon$ for all $\mathcal{A}$ with $|\mathcal{A}| \leqslant C$. Note that we have define the constant $\Lambda := \max_{\mathcal{A}:|\mathcal{A}|\leq C} 2\sigma(\mathcal{A},T)^2/(m-2) + 2Var(S_\tau)(m-1)/(m-2) + 2a\epsilon/3$. ∎

(a) Influence vs. time      (b) Error vs. #samples      (c) Error vs. #labels

Figure 5: On the **random** kronecker networks with 1,024 nodes and 2,048 edges, panels show (a) the estimated influence with increasing time window $T$; (b) the average relative error for different number of samples, each of which has 5 random labels for every node; and (c) the average relative error for varying number of random labels assigned to every node in each of 10,000 samples. For both (b) and (c), we set $T = 10$.

(a) Influence vs. time      (b) Error vs. #samples      (c) Error vs. #labels

Figure 6: On the **hierarchical** kronecker networks with 1,024 nodes and 2,048 edges, panels show (a) the estimated influence with increasing time window $T$; (b) the average relative error for different number of samples, each of which has 5 random labels for every node; and (c) the average relative error for varying number of random labels assigned to every node in each of 10,000 samples. For both (b) and (c), we set $T = 10$.

## F    Additional Experimental Results

In this section, we report additional experimental results on accuracy of influence estimation, continuous-time influence maximization and scalability for the synthetic networks.

### F.1    Accuracy of Influence Estimation

Figure 5 evaluates the estimated scope of influence for different time windows and the relative errors with respective to different number of random samples and labels on the random kronecker networks with 1,024 nodes and 2,048 edges. Figure 6 further reports similar results on the hierarchical kronecker networks. In all cases, the errors decrease dramatically as we draw more samples and labels.

In addition, because INFLUMAX can produce exact closed form influence on sparse small networks with exponential transmission functions, we compare CONTINEST with INFLUMAX in Figure 7, where we chose the highest degree node in the network as the source. We have drawn 10,000 random samples, each of which has 5 random labels for each node. CONTINEST outputs values of influence which are very close to the exact values given by INFLUMAX, with relative error less than 0.01 in all three types of networks.

Figure 7: Infected neighborhood size over three different types of networks with the exponential transmission function associated with each edge. Each type of network consists of 128 nodes and 141 edges. For panels (d-i), we set the observation window $T = 10$.

Figure 8: Panels present the influence against the number of sources by $T = 5$ on the networks having 1,024 nodes and 2,048 edges with heterogeneous Weibull transmission functions.

Figure 9: Panels present the influence against the time window $T$ using 50 sources on the networks having 1,024 nodes and 2,048 edges with heterogeneous Weibull transmission functions.

## F.2 Continuous-time Influence Maximization

We compare CONTINEST to other influence maximization methods based on discrete-time diffusion models: traditional greedy [1], with discrete-time Linear Threshold Model (LT) and Independent Cascade Model (IC) diffusion models, and the heuristic methods SP1M [2] and PMIA [25]. For INFLUMAX, since it only supports exponential pairwise transmission functions, we fit an exponential distribution per edge. Furthermore, INFLUMAX is not scalable; when the average network density of the synthetic networks is $\sim 2.0$, the run time for INFLUMAX is more than 24 hours. Instead, we present the results of CONTINEST using fitted exponential distributions (Exp). For the discrete-time IC model, we learn the infection probability within time window $T$ using Netrapalli's method [24]. The learned pairwise infection probabilities are also served for SP1M and PMIA, which essentially approximately calculate the influence based on the IC model. For the discrete-time LT model, we set the weight of each incoming edge to a node $u$ to the inverse of its in-degree, as in previous work [1], and choose each node's threshold uniformly at random. Figure 8 compares the expected number of infected nodes against source set size for different methods. CONTINEST outperforms the rest, and the competitive advantage becomes more dramatic the larger the source set grows. Figure 9 shows

Figure 10: Panels(a-b) show the running time against the network density by fixing the number of sources at 10 on the random and hierarchal kronecker network with 128 nodes. Panels(c-d) present the running time as we increase the number of selected source nodes on the networks with 128 nodes and 256 edges.

the expected number of infected nodes against the time window for 50 selected sources. Again, CONTINEST performs the best for all three types of networks.

## F.3 Scalability

Figure 10 compares CONTINEST to INFLUMAX and the Naive Simulation (NS) method in terms of running time for the continuous-time influence maximization problem over the random and hierarchal kronecker type of networks, respectively, with different densities and sizes on a single 2.4Ghz CPU core. For CONTINEST, we have drawn 10,000 samples, each of which has 5 random labels assigned to each node. For NS, we follow the work [1] to run 10,000 Monte Carlo simulations. For running times longer than 24 hours, we use dashed line to qualitatively indicate the estimated performance based on the time complexity of each method.