[Reviews · NeurIPS 2013]

Submitted by Assigned_Reviewer_2

Problem: Given a directed graph G compute the subset of k vertices whose infections leads to the maximum number of nodes being infected before time T, under the independent cascade model.

The paper builds heavily on top of the original paper on influence maximization in continuous-time diffusion networks (Gomez-Rodriguez & Scholkopf, ICML 2012). There are two pieces of cleverness that appear to be unique to this paper:

I. Reformulate continuous-time diffusion networks in terms of independent variates on the edges.

The basic contribution of this paper seems to stem from the observation that a continuous-time diffusion network can be converted from a distribution over the contagion times for nodes (t_i) to a distribution over transmission times between nodes (\tau_{ji}). Under the independent cascade model, the transmission times between nodes are independent.

The independent cascade model is a strong assumption in practice, but one the authors acknowledge freely. Have the authors considered whether there are natural relaxations of the independent cascade model which do not violate the independence of transmission times too much: e.g., allow a node to be more receptive to contagion if a large number of nodes, not necessarily direct parents, are contaminated. Such a relaxation might help capture the effects of mass advertising in the viral marketing problem.

II. Estimating the objective using Cohen's algorithm

The second contribution of this paper is to show that the objective can be reformulated as a neighborhood size estimation problem, and that Cohen's randomized algorithm can be directly applied to estimate the objective given sample data (Section 5).

I was surprised that the authors did not mention that Cohen's algorithm has been used before for influence maximization, in a different setting (Chen et al, KDD 2009 : Section 2). The authors cite the Chen paper in a laundry list of papers which consider influence maximization, but neglect to mention how Chen et al. use Cohen's algorithm to compute the transitive closure of a directed graph to speed up a submodular influence maximization problem under a weighted cascade model. I'll point out that Cohen's algorithm handles both transitive closure and neighborhood counting on a directed graph.

I understand that continuous-time diffusion networks are a different model from that of Chen et al., but its use in this paper is so similar to that of Chen et al., it begs the question---is Section 5 a totally novel use of Cohen's algorithm; or is it a novel reworking of an idea used previously for influence maximization, to continuous-time diffusion networks?

General Comments:

The paper does not assume too much in the way of prior knowledge: the continuous-time independent cascade model is explained (Section 2) as is submodular influence maximization (Section 6). The material should be accessible to a broad audience.

The experimental results are quite compelling: ConTinEst is vastly faster than the alternatives, and handles more general models of transition time.

Minor comments:

L671: Typo "Berstein's"
Summary: Compelling results, elegant algorithm. I have concerns that the authors haven't appropriated credited Chen et al. (KDD 2009), who appears to have used very similar ideas for a very similar problem, to a very similar end.

Submitted by Assigned_Reviewer_4

This paper presents a graphical formulation of continuous time diffusion model, a randomized algorithm (adapted from Cohen 1997) to perform efficient approximate inference, and optimizing a sub-modular function over the set of source nodes for influence maximization. Results on synthetic (Kronecker graph) data sets and real, Member tracker datasets are reported, estimation accuracy and run time are reported against baselines of earlier work infmax/ICML'12, inference with naive sampling strategy, and a few alternatives in [Netrapalli, SIGMETRICS'12].

This paper is attacking a significant problem, have novel insights that leads to efficient and effective algorithms. Compelling results are reported on datasets of significant size against a range of baselines methods, in both the main text and supplemental material. The paper is dense, but quite well-written.

The shortest path property (mentioned in page 2 Sec 2 and used in Sec 3 and 4) leads to the most likely diffusion path (e.g. A-->B directly, where A and B are some nodes), and not averaged over all possible transmission paths (e.g. averaging with A-->C-->B). The authors did not state this in the text, but I felt this may not be too apparent to mention.

It would be nice to see if the most influential nodes inferred from this algorithm concur with intuition and/or social network insight here (e.g. for the meme tracker data).
Summary: This paper is attacking a significant problem, have novel insights that leads to efficient and effective algorithms. Compelling results are reported on datasets of significant size against a range of baselines methods, in both the main text and supplemental material.

Submitted by Assigned_Reviewer_5

The paper addresses how to estimate and maximize influence in large networks, where influence of node (or set of nodes) A is the expected number of nodes that will eventually adopt a certain idea following the initial adoption by A. The authors develop an algorithm for estimating influence within a given time frame, then use it as the basis of a greedy algorithm to find a given number of nodes to (approximately) maximize influence within the given time frame. They present theoretical bounds and an experimental evaluation of the algorithm.

The authors build on an extensive list of existing work, which is appropriately cited. The most relevant is the work by Gomez-Rodriguez & Scholkopf (2012), which provides an exact analytical solution to the identical formulation of the influence estimation problem. The main innovation in the present paper is a fast randomized algorithm for estimating influence, which is based on the algorithm for estimating neighborhood size by Cohen (1997). This approximation allows more flexibility in modeling the flows through the edges, is substantially faster than the analytical solution, and scales well with network size. Overall, this is a solid paper on an important topic of practical relevance.

I am not sure I can agree with the authors' conclusion that "Clearly, CONTINEST estimates the real influence of each source node significantly better than competitive models." In Figure 3a, the differences in Mean Absolute Error between the best and the worst method are about 2, and the differences between the best and the second-best method are about 1. I am assuming that the unit of measurement is "nodes". In a network of 600 nodes, 1-2 nodes seem to be small difference (but it is difficult to judge this without knowing the levels of influence in the dataset and variance in error, which are not reported). Also, the use of 'significantly' is best reserved for differences that are statistically significant.

It would be useful to have: a verbal explanation of Equation 6; run times of the various algorithms on the real-world network; a measure of variance in Figure 3.

The supplementary material contains quite a bit of useful information. I suggest moving some of the experimental results to the main paper, especially section F2. This should be possible in terms of space. While the paper is reasonably clear, it can be written more succinctly and elegantly without losing content.
Summary: This is a solid paper on estimating influence in large networks, an important topic of practical relevance.
Author Feedback

Author rebuttal: We first thank the reviewers for their constructive and valuable comments. We want to emphasize the main contributions of the paper:

1. We proposed a novel graphical model view of the influence estimation problem in continuous-time diffusion networks, which enables us to reduce the problem to a sequence of parallelizable neighborhood estimation problems in static graphs.

2. We proposed an efficient algorithm for influence estimation and maximization in continuous-time diffusion networks with provable performance guarantees.

3. Experiments on both large scale synthetic networks (up to one million nodes) and real network data (memetracker) show that our algorithm improves over previous state-of-the-arts in terms of the accuracy of the estimated influence, the quality of the selected nodes in maximizing the influence and the run-time of reaching the solution.

******Reviewer 2:

For continuous-time models that deviate from the assumption of independent cascades, one can first approximate the model locally with independent cascade models. Due to the graphical model view laid out in the paper, such approximation can be well justified from a graphical model variational inference point of view.

Our algorithm is a novel use of Cohen's algorithm, and it is substantially different from that of Chen et al. KDD 2009 in the following key aspects:

1. The transitive closure problem in Chen et al is quite different from the neighborhood-at-distance-T problem in our algorithm. The former can be viewed as a special case of the latter where the distance is fixed but set to a very large number. Furthermore, in the latter problem, our algorithm needs to be designed in such a way that queries for a range of distance values T can be carried efficiently. Therefore, additional algorithmic steps (modified Dijkstra's algorithm) and special data structure (least-label list) need to be constructed to support such queries. Although both Chen et al. and us used the exponential estimator as in equation (9), the similarity stops there. The modified Dijkstra's algorithm and least-label-list components are not used by Chen et al.

2. Our second novel use of the Cohen algorithm arises in the case of influence estimation for multiple sources. Again, the computation is drastically reduced by cleverly using the least label least obtained from single source case, which is not considered at all by Chen et al.

3. We have provided formal statistical guarantees for our influence estimation and maximization algorithms using Cohen's algorithm as a component. Such formal guarantees are missed in the paper of Chen et al.

We will clarify these difference in the final version of the paper.

******Reviewer 4:

We note that the shortest path property is applicable to the case when the set of transmission times (edge weights) are fixed. In our algorithm, we need to sample many sets of transmission times, and the shortest paths may change from A-C-B to A-B depending on the sample. Thus, our algorithm can take into account the effects of both A-B and A-C-B being potential shortest paths and "average" the results.

We investigated the web sites corresponding to the selected 10 sources which maximize the influence within a month in the memetracker dataset. They include digg.com (a popular news site), lxer.com (Linux news and open sources), exopolitics.blogs.com (politics), mac.softpedia.com (mac rumors), gettheflick.blogspot.com (pictures), urbanplanet.org (the world's favorite discussion forum for urban enthusiasts), givemeaning.blogspot.com (politics), talkgreen.ca (environmental protection), curriki.org (education), and pcworld.com (technology). This set consists of highly popular sites and other less popular sites. While this is conforming to the common intuition of viral marketing, we emphasize that our algorithm is also able to improve quantitatively the actual influence.


******Reviewer 5:

In Figure 3(a), we actually reported the standard error at each data point by randomly splitting all the cascades into a 80% training set and a 20% test set and repeating the random splitting for five times in total. The average standard error for ConTinEst is 0.0024, which is too small for the error bar to be obvious in the Figure 3(a).

The reason for the 1-2 nodes difference in the performance of different algorithms is that the length of the observed cascades roughly conforms to a power-law distribution where most cascades are very short consisting normally of 2-4 nodes while few of them are relatively long. The difference of the estimation error is thus relatively small, but statistically significant (we can add formally statistical test results).

We emphasize that such improvement in the estimation accuracy for each individual node is critical for choosing nodes to maximize longer-term influence. In this case, roughly the overall diffusion model needs to "pieces" or "chains" together the influence of individual nodes, and the error in the influence estimation for individuals will accumulate along the diffusion paths. Hence any consistent improvement in influence estimation for individual nodes can lead to significant improvement in overall influence estimation and maximization process.